# Exploring Food Deserts in Seoul, South Korea during the COVID-19 Pandemic (from 2019 to 2021)

**Jeon-Young Kang** [1] and **Seunghwan Lee** [2,*]

1   Department of Geography Education, Kongju National University, Gongju 32588, Korea;
    geokang@kongju.ac.kr
2   Department of Tourism Management, Kongju National University, Gongju 32588, Korea
*   Correspondence: slee@kongju.ac.kr; Tel.: +82-41-850-8675

**Abstract:** Since the coronavirus disease 2019 (COVID-19) was declared a pandemic by the World Health Organization, our lifestyle (e.g., food culture) has changed. In particular, the food insecurity issue has exacerbated. To address this issue, this study aims to measure spatial accessibility to food outlets and identify food deserts in Seoul, South Korea during the COVID-19 pandemic (i.e., 2019–2021). To assess spatial access to food outlets, we used the enhanced two-step floating catchment area (E2SFCA) method. The results from the E2SFCA methods showed that spatial accessibility to restaurants increased, but access to grocery stores decreased. A noticeable change occurred in Gangnam and Seocho. The Gini coefficients indicated that equality in spatial accessibility to restaurants fluctuated (i.e., worsened from 2019 to 2020 and improved from 2020 to 2021), whereas equality in spatial accessibility to grocery stores improved. The results help to identify prioritized regions where additional food resources can be placed, especially for marginalized people who have limited access to food due to their socio-economic status.

**Keywords:** food desert; spatial accessibility; COVID-19; two-step floating catchment area method; GIS

## 1. Introduction

Since the World Health Organization (WHO) declared the coronavirus disease 2019 (COVID-19) pandemic, many countries have experienced the subsequent spread of COVID-19 and loss of life. The COVID-19 pandemic has changed our lives in many ways, including the implementation of lockdowns and remote work [1]. In recent years, many countries have slowly allowed people to go back to their workplaces, but some still adhere to strict social distancing protocols [2].

The COVID-19 pandemic has also shaped food culture and industries. According to the Seoul Institute [3], the foodservice industry was strongly affected in Seoul, South Korea. People were more likely to stay at home and not go outside due to their concerns about COVID-19 infection [4]. Under this situation, small food service businesses closed, which may have been responsible for the limited provision of food resources to people. Food deserts, which refer to places with limited food sources within a certain distance [5], became more common during the pandemic [6,7]. Therefore, it is important to provide a better understanding of people's accessibility to food resources.

Spatial accessibility, which assesses people's accessibility to a particular public service, has long been an interest in many fields, such as urban planning [8], social work [9], public health [10,11], and geography [12,13]. Even during the COVID-19 pandemic, spatial accessibility to COVID-19-related healthcare resources (e.g., testing sites, ventilators) was measured [14–16]. As measuring spatial accessibility helps identify where additional services (or resources) need to be placed [13], it was necessary to assess people's spatial accessibility to food outlets during the COVID-19 pandemic.

To address these issues, this study mainly focuses on measuring spatial accessibility to grocery stores and restaurants in Seoul, South Korea. The spatial accessibility to food outlets

is assessed using the enhanced two-step floating catchment area (E2SFCA) method. Our research questions are as follows. (1) Did spatial accessibility to grocery stores change for Seoul citizens during the peak of the COVID-19 pandemic? (2) Did spatial accessibility to restaurants change for Seoul citizens during the peak of the COVID-19 pandemic? (3) Were there fluctuations in the spatial accessibility to grocery stores and restaurants in Seoul during the peak of the COVID-19 pandemic?

## 2. Methods

### 2.1. Study Area and Data

This study focused on spatial accessibility to food outlets in Seoul, South Korea. The capital city of South Korea, Seoul, is highly urbanized and is home to about 10 million people. To measure spatial accessibility, we used three types of datasets. First, we used the daytime (9 a.m.–6 p.m.) population data provided by the Seoul Data Portal (https://data.seoul.go.kr) (accessed on 2 January 2022). The population size was estimated based on the national survey and mobile phone usage [17]. The Seoul Data Portal provides the estimated population at each hour at the k-unit level. The k-unit is the minimum level of spatial units of the socio-economic data provided by the Korean government. In our analysis, we used daytime population data in March 2019, March 2020, and March 2021 and considered only weekdays (Figure 1). Since the Korean government initiated the social distancing plan in March 2020, it is appropriate to take March 2019, 2020, and 2021 as the study period for assessing to what extent spatial accessibility has been changed over time in the era of COVID-19.

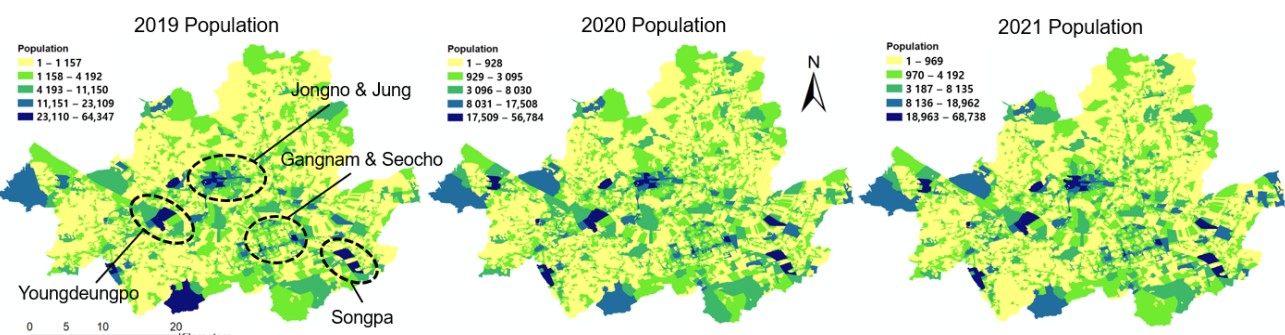

**Figure 1.** Population in Seoul from 2019 to 2021.

As shown in Figure 1, the regions (i.e., Jongno, Jung, Gangnam, Seocho, Songpa, and Youngdeungpo) highlighted in dark represent the business districts in Seoul. These regions have relatively higher populations during the daytime. There were not many differences in the spatial distribution of the daytime population from 2019 to 2021.

Second, we used food outlet data obtained from the Korean Data Portal (https://www.data.go.kr) (accessed on 2 January 2022). We categorized food outlets into restaurants and grocery stores. The restaurants denoted any type of restaurant, including delivery-only restaurants and fast foods. The grocery stores included wholesale stores, convenience stores, and supermarkets, among others. Table 1 presents the number of restaurants and grocery stores for 2019, 2020, and 2021. The number of restaurants gradually increased in 2019, and the number of grocery stores dramatically decreased in 2021. Table 2 provides the number of food outlets by total population from 2019 to 2020. Despite an increase in the numbers of restaurant and grocery stores, the proportion of food outlets to the total population has not changed much.

**Table 1.** Number of Grocery Stores and Restaurants from 2019 to 2021.

| Date | All Businesses | Restaurants | Grocery Stores |
|---|---|---|---|
| March 2019 | 382,245 | 54,282 (14.20%) | 17,410 (4.55%) |
| March 2020 | 391,500 | 55,122 (14.08%) | 17,608 (4.50%) |
| March 2021 | 315,555 | 55,308 (17.53%) | 14,997 (4.75%) |

**Table 2.** The Number of Food Outlets by Population from 2019 to 2021.

| Date | All Businesses by Total Population | Restaurants by Total Population | Grocery Stores by Total Population |
|---|---|---|---|
| March 2019 | 0.0337 | 0.0048 | 0.0015 |
| March 2020 | 0.0354 | 0.0050 | 0.0016 |
| March 2021 | 0.0291 | 0.0051 | 0.0014 |

Figure 2 illustrates the spatial distributions of restaurants and grocery stores in Seoul from 2019 to 2021. In general, more than 5000 restaurants and 1000 grocery stores are located in Gangnam. Other regions had similar numbers of restaurants and grocery stores.

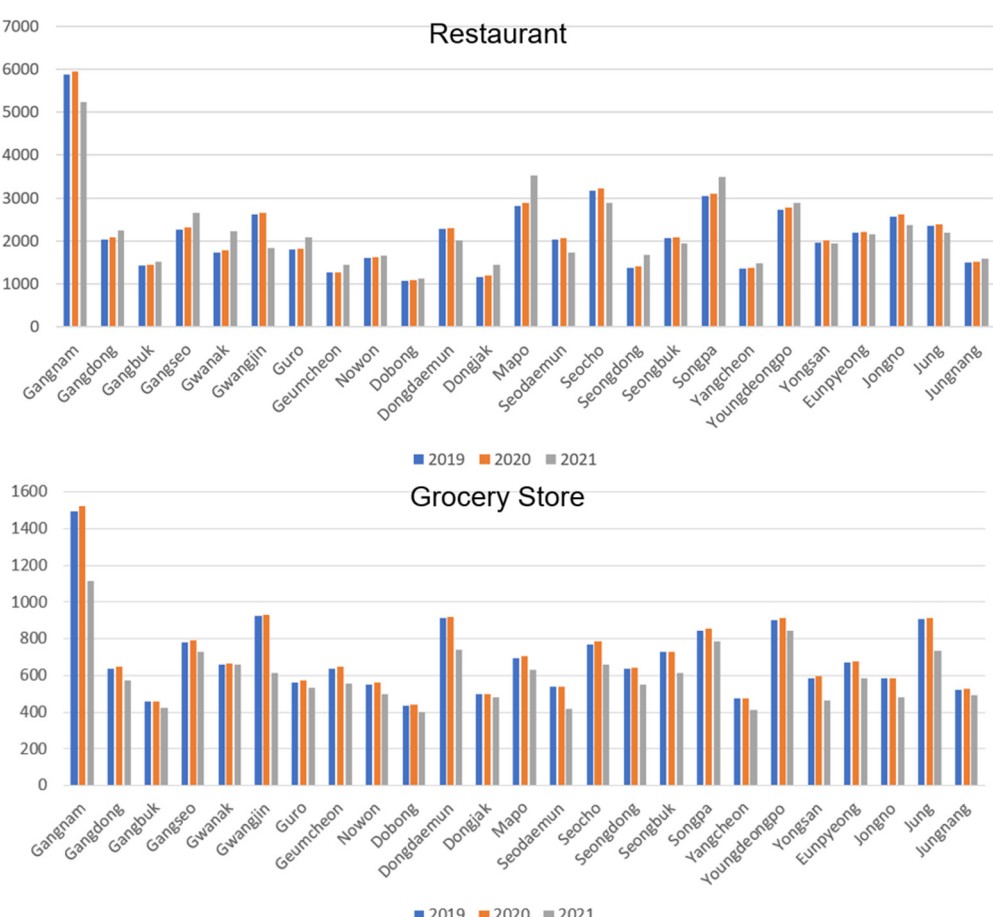

**Figure 2.** Spatial distributions of restaurants and grocery stores.

Third, we used road network data to calculate travel time from the population center to food outlets and vice versa. As we considered delivery, stop-by, and to-go services, we used road network data for driving and walking. Using a Python library, OSMnx (https://osmnx.readthedocs.io/en.stables/) (accessed on 2 January 2022)., we obtained road network data. OSMnx not only provided the road network data for free but also analyzed catchment areas for the urban infrastructures.

### 2.2. E2SFCA Method

The 2SFCA method is used to measure spatial accessibility based on the catchment areas of the testing sites and population centers (i.e., centroids of each administrative unit) with the number of populations [18]. The catchment areas of each testing site and population center can be delineated based on the length of road segments and travel time, which can be formed as a polygon. When the population center reaches the catchment area, the number of populations at the population center is considered the magnitude of the demand for testing sites. The catchment areas overlap when multiple testing sites are closely distributed. We summed up the values of the overlapping regions. Therefore, highly overlapping regions generally have higher accessibility scores.

This study used the E2SFCA method [10], which is a modified version of the 2SFCA method. The E2SFCA method advances the previous 2SFCA method in that it can incorporate distance decay, which implies that people may prefer to visit closer service locations than farther locations. The E2SFCA method has two steps.

First, the ratio of supply to demand in the service area is measured (1). In this step, the numbers of populations at k within the catchment areas (i.e., $t_1$, $t_2$, and $t_3$) of the service location are summed up based on the weights (i.e., $W_1$, $W_2$, $W_3$). The weights are assigned based on the distance decay. Then, the supply at a specific location is divided by the total population within the catchment area.

$$R_j = \frac{S_j}{\sum_{k\epsilon(t_{kj}<t_1)} P_k W_1 + \sum_{k\epsilon(t_1<t_{kj}<t_2)} P_k W_2 + \sum_{k\epsilon(t_2<t_{kj}<t_3)} P_k W_3} \tag{1}$$

where $S_j$ is the supply at each service location ($j$), $P_k$ is the number of populations at location $k$, and $W$ (i.e., $W_1$, $W_2$, and $W_3$) is the weight based on the distance from population location $k$ to service location $j$ ($t_{kj}$). As this paper focuses on the availability of testing, $S_j$ is equal to 1.

The E2SFCA method incorporates three distance intervals for the catchment area. The weights are determined according to the Gaussian curve [10]. In the case of driving, we assumed that people were willing to visit the testing sites for up to 30 min. The weights are as follows: (1) one value for 0–10 min, (2) 0.68 for 10–20 min, and (3) 0.22 for 20–30 min. In the case of walking, we assumed that people were willing to visit the testing sites for up to 15 min. The weights were as follows: (1) 1 for 0–5 min, (2) 0.68 for 5–10 min, and (3) 0.22 for 10–15 min.

Second, the ratio of demand to supply at location i is summed up based on the maximum travel times (driving: 30 min; walking: 15 min). The weights are also applied in this step.

$$A_i^M = \sum_{j\epsilon(t_{ij}<t_1)} P_j W_1 + \sum_{j\epsilon(<t_1<t_{ij}<t_2)} P_j W_2 + \sum_{j\epsilon(t_2<t_{ij}<t_3)} P_j W_3 \tag{2}$$

where $A_i^M$ is the spatial accessibility at location i when using a specific transportation mode (i.e., driving or walking), and $R_j$ is the value calculated by Equation (1), which signifies the ratio of supply to demand. The weights ($W_1$, $W_2$, and $W_3$) denote 1, 0.68, and 0.22, respectively.

### 2.3. Inequality Measurement

To determine whether there was inequality in accessing restaurants and grocery stores, we used the Gini coefficient, which is a well-known measure of the concentration of values. In the literature, the Gini coefficient has been widely used to assess inequality towards various public services [13,16]. The Gini coefficient ranged from zero to one. Zero means that the accessibility scores are evenly distributed across a space, which implies that all people have the same ease in terms of geographical access to restaurants and grocery stores. Conversely, one indicates a high concentration of higher (or lower) accessibility scores. In this case, it is easy (or difficult) for people who reside in some regions to access restaurants and grocery stores. When the Gini coefficient is measured for spatial accessibility over

time, it is helpful to understand to what extent spatial accessibility to a specific service has improved (or worsened).

## 3. Results

Table 3 and Figure 3 present the results of the E2SFCA method. To answer the research questions about the extent to which the measures of spatial accessibility to restaurants vary over time, we performed a one-way analysis of variance (ANOVA) test. Significant differences were found in the spatial accessibility scores of restaurants between 2019, 2020, and 2021: $F_{2,57,456} = 59.99$, $p < 0.001$. The results from the Bonferroni adjustment showed that the spatial accessibility measures in 2019 were greater than those in 2021 ($p < 0.001$) and that the measures in 2020 were greater than those in 2021 ($p < 0.001$).

**Table 3.** Accessibility Scores.

| Type | Year | Average (IQR) |
|---|---|---|
| Restaurant | 2019 | 0.0092 (0.0051) |
| | 2020 | 0.0094 (0.0052) |
| | 2021 | 0.0101 (0.0054) |
| Grocery | 2019 | 0.0033 (0.0781) |
| | 2020 | 0.0033 (0.0797) |
| | 2021 | 0.0030 (0.0697) |

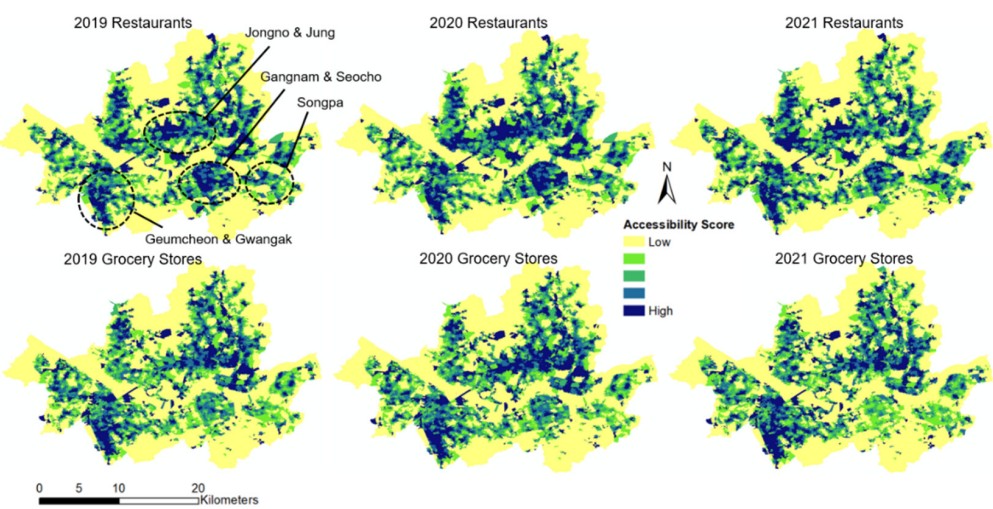

**Figure 3.** Spatial accessibility to restaurants and grocery stores (2019–2021).

A one-way ANOVA test was also used to determine the differences in spatial accessibility of grocery stores between 2019, 2020, and 2021, and statistical differences were found: $F_{2,57,456} = 48.98$, $p < 0.001$. The results from the Bonferroni adjustment showed that the accessibility measures in 2019 were greater than those in 2021 ($p < 0.001$) and that the measures in 2020 were greater than those in 2021 ($p < 0.001$).

The Gini coefficients were used to determine whether there was inequality in spatial accessibility to restaurants and grocery stores (Table 4). In terms of spatial accessibility to restaurants, inequality worsened from 2019 to 2020 but improved from 2020 to 2021. Conversely, inequality in spatial accessibility to grocery stores has improved over the years.

**Table 4.** Gini Coefficients.

| Type | Year | Gini Coefficient |
|---|---|---|
| Restaurants | 2019 | 0.2770 |
| | 2020 | 0.2832 |
| | 2021 | 0.2640 |
| Grocery Stores | 2019 | 0.2974 |
| | 2020 | 0.2804 |
| | 2021 | 0.2685 |

## 4. Concluding Discussion

Since March 2020, the unforeseen and unfortunate occurrence of COVID-19 has resulted in massive financial loss and economic crisis worldwide. Specifically, this event seriously affected small businesses, including restaurants and grocery stores. To address this issue, we aimed to seek food deserts, which relate to the degree of spatial accessibility of food service-related stores, in Seoul, South Korea. Spatial accessibility to food showed the status of restaurants and grocery stores during the COVID-19 pandemic from 2019 to 2021.

According to the results, the accessibility scores looked different between restaurants and grocery stores. This is because these two industries have experienced different circumstances. The changes in accessibility scores during the peak of the COVID-19 pandemic may be due to the increased number of restaurants rather than a population change. This result seems ironic because the last two years have been an economically challenging period. Prior research [3,19] has presented supporting claims to explain this phenomenon. First, restaurant owners did not want to close their stores because they hoped that the pandemic would end soon. Second, restaurants provided takeout and delivery services to overcome this difficulty. Third, unemployed people attempted to open new restaurants, which had a comparably low entry barrier. Fourth, employers could not fire their employees because of the Labor Standards Act. Thus, they would close the store and then reopen it as a family business.

These explanations can also be used to describe the different patterns between the central business district (CBD) and the residential district. For the CBD, such as the Gangnam and Jongno areas, accessibility scores tended to be higher in 2020 than in 2019. After a year, the score was lower in 2021 than in 2020. As many news articles have stated, due to the escalation of the pandemic in 2020, many companies strongly recommended that their workers work remotely from home. Thus, the population in the CBD decreased. This situation increased the accessibility score. However, in March 2021, the status of the COVID-19 pandemic changed slightly. As the pandemic seemed to slow down, many employees started to commute again. Thus, the accessibility score in the CBD decreased. Second, the score of the residential district showed an opposite pattern to that of the CBD. For example, in the Songpa area, the score did not change much from 2019 to 2020, but it became higher in 2021 because the number of restaurants increased while the population decreased in this area. This is due to the fact that a year after the COVID-19 outbreak, telecommuters returned to the workplace again, and more restaurants opened in the Songpa area at the same time.

This study also analyzed the accessibility scores of grocery stores. In March 2019 and March 2020, the scores seemed identical, although the 2021 score was lower than that of previous years. As described in the graph, the number of grocery stores in 2021 was less than in the previous year. According to a news article [20], some hypermarkets in Korea shut down temporarily or closed forever due to COVID-19. This happened because the continuing social distance policy deterred people from visiting supermarkets due to the rapid growth of online stores during the peak of COVID-19 pandemic. Specifically, the changes in the Gangnam and Jongno areas between 2020 and 2021 were the most notable. However, contrary to this trend, the score in the Songpa area, which is a residential district, was slightly higher because of the presence of supermarkets in the community. This means that the older generation was likely to visit the market because of their unfamiliarity with online shopping.

This study measured the Gini coefficients to account for equality of spatial accessibility of the food service-related industry during the pandemic. According to previous studies [21,22], the Gini coefficient is generally negatively perceived when greater than 0.4, but all the numbers indicated in this study were lower than this threshold. Based on the results, during the peak of the COVID-19 pandemic, positive spatial accessibility was indicated in Seoul metropolitan city because there were sufficient restaurants and grocery stores, meaning the COVID-19 situation did not seem to cause food deserts in this large city. However, despite this result, the coefficients between restaurants and grocery stores showed slight differences between 2019 and 2020 because people visited these two sectors for different purposes. People always go to the grocery store, and when they do, they are likely to stay for a short time. However, people tend to cut down on their restaurant visits whenever necessary, such as during the peak of the COVID-19 pandemic.

## 5. Implications and Limitations

This study has some implications based on the results. First, this study is meaningful in that a food desert study was conducted during a specific unexpected period. During this challenging era, food is the most necessary resource. Therefore, measuring the spatial accessibility of food is a timely piece of research at this moment in time. In addition, this study measured the spatial accessibility of groceries and restaurants, unlike previous studies, e.g., [18,23,24], that examined the spatial accessibility of grocery retailers or fast food restaurants. However, by adding restaurants to the measurement of food deserts in this study, the result could account for a more precise status of food-related small businesses. Second, the results provide important practical implications. Previous food desert studies, e.g., [25], have provided implications for marginalized people who have limited access to food. Conversely, the results of the present study showed that, in terms of food deserts during unexpected periods, such as the peak of the COVID-19 pandemic, a different perspective is needed and that small business owners should be considered. According to the results, as expected, food deserts were not observed in urban areas during this challenging era. This was due to small business owners having a tough time continuing their businesses and being unable to close them. Moreover, many unemployed individuals attempted to open a food service business because of its comparably low entry barrier. For these reasons, this study cautiously recommends that the government provide appropriate support for small business owners.

This study has several limitations. First, we did not consider temporal dynamics (i.e., traffic and supply and demand) as spatial accessibility could be temporally dynamic [26]. Second, as people could go to food outlets using multiple transportation modes [11], spatial accessibility through cycling or public transportation should have been measured. Finally, the measurement would have been more accurate had we measured spatial accessibility using empirical data on the extent to which people were willing to spend time acquiring particular types of foods.

**Author Contributions:** Conceptualization, data analysis, visualization, and writing: J.-Y.K.; concluding, review, and writing: S.L. All authors have read and agreed to the published version of the manuscript.

**Funding:** This research received no external funding.

**Institutional Review Board Statement:** Not applicable.

**Informed Consent Statement:** Not applicable.

**Data Availability Statement:** All datasets are available from the corresponding authors by reasonable request.

**Conflicts of Interest:** The authors declare no conflict of interest.

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
