# Peer review of "Exploring Food Deserts in Seoul, South Korea during the COVID-19 Pandemic (from 2019 to 2021)"

_sustainability, doi:10.3390/su14095210_

Round 1

Reviewer 1 Report

The paper determined access to food outlets in South Korea. A nice paper but needs improvement. Suggestions below

Title: I think the authors need to include the time frame/period of their study (e.g 2020 to 2021) in the paper title

Abstract: Rephrase line 8-9: Since the coronavirus disease-19 (COVID-19) WAS DECLARED A PANDEMIC by the World Health Organization….

Line 30-31 “According to the Seoul Institute (2021), the foodservice industry was strongly affected in Seoul, South Korea” . A citation and reference is required after the above sentence.

In line 40 and some other places in the manuscript, the authors write on COVID-19 pandemic as if the world is already in the post-COVID-19 era. At least, the WHO has not said so. You may say “during the peak of COVID-19 pandemic”….

Lines 50-52: I think the research question number 3 is basically the sum of questions 1 and 2. The authors need to modify question 3. You may write “Are there fluctuations between the spatial accessibility to grocery stores and restaurants in Seoul during the peak of COVID-19 pandemic, between year xxx and yyyy”?

Study Area and Data:  Information provided on the study area (Line 55-56) is very scanty. Apart from the population density, the authors can provide the readership with data on the number and accessibility of the two main food outlets prior to the study period (2019-2021). I think this can help the readership make a better sense of the food security/accessibility situation in the “good old days”

In the last paragraph of the introduction, the authors mentioned that the aim of the study was to determine spatial accessibility to grocery stores and restaurants in Seoul, South Korea during the COVID-19 pandemic. If this is the case, what is the rational for including 2019 in the study period? You may recall that COVID-19 was first reported in December 2019 and was later declared a pandemic in March 2021.

Line 67-68: the authors stated that “No significant differences were found in the spatial distribution of the daytime 67 population from 2019 to 2021” . Is the above statement part of their findings/results? If, why not present it in the result section? If not, why such statement in the M&M section?

Line 77-78: Is this part of the results?

Line 138-140: Substantiate your claim about Gini coefficient with a citation and reference it accordingly.

For this reviwer, the major flaw in this paper is the inclusion of year 2019 in the study which focused on accessibility to food out lets during the peak of COVID-19 pandemic. Except the authors provide a valid justification for this, the comparison of the results of the findings across the years (2019-2021) and the discussions there in do not make much sense in my view.

Lines 208, 221: “[e.g 13]”, “[e.g 6, 24]” are not the standard ways of citation in scholarly articles.

Discussion: The first paragraph of the discussion is more of introduction and result, not discussion.

Rather than sandwiching the conclusion of the study in the discussion, the authors can highlight their major findings and recommendation as a conclusion in a separate section (section 5) or as the last paragraph in the discussion.

Reference

The references were not formatted according to MDPI style, the first letters in the names of some journals were written in small letters

Author Response

The paper determined access to food outlets in South Korea. A nice paper but needs improvement. Suggestions below

Comment 1: Title: I think the authors need to include the time frame/period of their study (e.g 2020 to 2021) in the paper title

Response 1: Thank you for your suggestion. We have added time frame of the study (from 2019 to 2021)

Comment 2: Abstract: Rephrase line 8-9: Since the coronavirus disease-19 (COVID-19) WAS DECLARED A PANDEMIC by the World Health Organization….

Response 2: Thank you for your comment. We have followed your suggestion as you stated.

Comment 3: Line 30-31 “According to the Seoul Institute (2021), the foodservice industry was strongly affected in Seoul, South Korea” . A citation and reference is required after the above sentence.

Response 3: Thank you for your comment. We have added citation and reference after the sentence you provided.

Comment 4: In line 40 and some other places in the manuscript, the authors write on COVID-19 pandemic as if the world is already in the post-COVID-19 era. At least, the WHO has not said so. You may say “during the peak of COVID-19 pandemic”….

Response 4: Thank you for your suggestion. We have accepted your suggestion. “during the peak of COVID-19 pandemic”….

Comment 5: Lines 50-52: I think the research question number 3 is basically the sum of questions 1 and 2. The authors need to modify question 3. You may write “Are there fluctuations between the spatial accessibility to grocery stores and restaurants in Seoul during the peak of COVID-19 pandemic, between year xxx and yyyy”?

Response 5: Thank you for your comment. We revised the sentence you mentioned.

Comment 6: Study Area and Data:  Information provided on the study area (Line 55-56) is very scanty. Apart from the population density, the authors can provide the readership with data on the number and accessibility of the two main food outlets prior to the study period (2019-2021). I think this can help the readership make a better sense of the food security/accessibility situation in the “good old days”

Response 6: Thank you for your suggestion. In this study, we focus on the fluctuation of the spatial accessibility to food outlets. So, we have measured the spatial accessibility to grocery stores and restaurants during the year 2019 to 2021. As we all understand, the first case of the COVID-19 occurred in 2020 in South Korea. We think the spatial accessibility measure in March 2019 may help the readers to have a sense of food security/accessibility situation. We have revised our objectives more clearly in the revised manuscript.

Comment 7: In the last paragraph of the introduction, the authors mentioned that the aim of the study was to determine spatial accessibility to grocery stores and restaurants in Seoul, South Korea during the COVID-19 pandemic. If this is the case, what is the rational for including 2019 in the study period? You may recall that COVID-19 was first reported in December 2019 and was later declared a pandemic in March 2021.

Response 7: Thank you for your comment. We have attempted to collect data of a specific month(March) in three years. Given the Korean government initiated the social distancing policy in March 2020, we decided to think of to what extent such social distancing policy in response to the COVID-19 outbreaks influence the grocery stores and restaurants. So, we compared spatial accessibility to the food outlets in March in three years. We have added this rationale in the revised manuscript.

Comment 8: Line 67-68: the authors stated that “No significant differences were found in the spatial distribution of the daytime 67 population from 2019 to 2021” . Is the above statement part of their findings/results? If, why not present it in the result section? If not, why such statement in the M&M section?

Response 8:  Thank you for your comment. This sentence is not the result. We mentioned the basic information which the figure1 presents. We have revised our manuscript.

Comment 9: Line 77-78: Is this part of the results?

Response 9: This sentence accounts for the data in table 1 which is the number of grocery stores and restaurants from 2019 and 2021.

Comment 10: Line 138-140: Substantiate your claim about Gini coefficient with a citation and reference it accordingly.

Response 10: Thank you for your suggestion. We have properly cited and added a reference.

Comment 11: For this reviwer, the major flaw in this paper is the inclusion of year 2019 in the study which focused on accessibility to food out lets during the peak of COVID-19 pandemic. Except the authors provide a valid justification for this, the comparison of the results of the findings across the years (2019-2021) and the discussions there in do not make much sense in my view.

Response 11: Given the Korean government initiated the social distancing policy in March 2020, we decided to think of to what extent such social distancing policy in response to the COVID-19 outbreaks influence the grocery stores and restaurants. We have revised the findings and discussion.

Comment 12: Lines 208, 221: “[e.g 13]”, “[e.g 6, 24]” are not the standard ways of citation in scholarly articles.

Response 12: We appreciate your comment and revised the way of citation in those parts.

Comment 13: Discussion: The first paragraph of the discussion is more of introduction and result, not discussion.

Response 13: Thank you for your comment. We have revised the first paragraph.

Comment 14: Rather than sandwiching the conclusion of the study in the discussion, the authors can highlight their major findings and recommendation as a conclusion in a separate section (section 5) or as the last paragraph in the discussion.

Response 14: Thank you for your comment. We have seperated conclusion section into 4.concluding discussion and 5.implication and limitation.

Comment 15: Reference

The references were not formatted according to MDPI style, the first letters in the names of some journals were written in small letters

Response 15: Thank you for your comment. We have properly changed the format.

Reviewer 2 Report

This article presented an important issue during the COVID 19 pandemic emphasizing small businesses or groceries. however, some questions need to be answered.

1. Why author collected and presented data for the month of March for consecutive three years only? What about other months or whole years?

2. 2SFCA method is an excellent method but authors should present the population size parallelly with the number of groceries or restaurants. 

3. I did not see any concluding remarks based on objective 3?

Author Response

This article presented an important issue during the COVID 19 pandemic emphasizing small businesses or groceries. however, some questions need to be answered.

Comment 1: Why author collected and presented data for the month of March for consecutive three years only? What about other months or whole years?

Response 1: We have attempted to collect data of a specific month(March) in three years. Given the Korean government initiated the social distancing policy in March 2020, we decided to think of to what extent such social distancing policy in response to the COVID-19 outbreaks influence the grocery stores and restaurants. So, we compared spatial accessibility to the food outlets in March in three years. We have added this rationale in the revised manuscript.

Comment 2: 2SFCA method is an excellent method but authors should present the population size parallelly with the number of groceries or restaurants. 

Response 2: Thank you for your comment.. We have added Table 2, which provides the number of food outlets by population from 2019 to 2021. 

Comment 3: I did not see any concluding remarks based on objective 3?

Response 3: Thank you for your comment. We attempted to address characteristics of two industries during the covid pandemic in terms of accessibility scores. However, followed by your comments, we added some sentences to indicate that those parts are accounting for the differences. 

Round 2

Reviewer 1 Report

The authors have significantly improved the scholarly quality/ publishable value of the manuscript.